# In Elderly Anemic Patients without Endoscopic Signs of Bleeding Are Duodenal Biopsies Always Necessary to Rule Out Celiac Disease?

**DOI:** 10.3390/diagnostics12030678

**Published:** 2022-03-10

**Authors:** Giulia Pivetta, Chiara Coluccio, Emanuele Dilaghi, Edith Lahner, Emanuela Pilozzi, Marilia Carabotti, Vito Domenico Corleto

**Affiliations:** 1Department of Medical-Surgical Sciences and Translational Medicine, Sant’Andrea Hospital, University Sapienza, 00189 Rome, Italy; giulia_pivetta@libero.it (G.P.); colucciochiara@gmail.com (C.C.); emanuele.dilaghi@uniroma1.it (E.D.); edith.lahner@uniroma1.it (E.L.); marilia.carabotti@uniroma1.it (M.C.); 2Department of Clinical and Molecular Medicine, Sant’Andrea Hospital, University Sapienza, 00166 Rome, Italy; emanuela.pilozzi@uniroma1.it

**Keywords:** anemia, celiac disease, duodenal biopsies

## Abstract

Iron-deficiency anemia in the elderly may be due to numerous gastrointestinal conditions. Anemia is frequent in celiac disease (CD); however, the use of routine duodenal biopsies, independently of age or serology, is debated. To determine the diagnostic yield of routine duodenal biopsies in adult and elderly patients with no bleeding anemia, a cross-sectional study analyzing 7968 gastroscopies (2017–2020) was performed; 744 were for anemia and 275 were excluded (GI bleeding or without duodenal biopsies). Of the 469 included patients, clinical, endoscopic, and histological features were analyzed in groups with or without histopathological changes in the duodenal mucosa (DM). Univariate/multivariate analyses were performed. Of the 469 patients, 41 (8.7%) had DM histopathological changes, 12 (2.6%) had CD, 26 (5.5%) had duodenal intraepithelial lymphocytosis (DIL), and 3 had (0.6%) other conditions. They were younger compared to patients with normal DM. DM histopathology was significantly inversely correlated with age group, with prevalences of 27%, 20%, 12.5%, 10%, and 2.5%, in the <40–50, 51–60, 61–70, 71–80, and >80-year age groups, respectively (*p* = 0.0010). Logistic-regression models showed that anemic patients aged >60, >70, or >80 years with endoscopically normal DM had a progressively three- to four-fold higher probability of having normal duodenal histology. In adults, anemic patients without bleeding, age and endoscopically normal DM are predictors of normal DM histology. In >70-year anemic patients, negligible DM pathology was found. The results suggest that routine duodenal biopsies are questionable in elderly anemic patients

## 1. Introduction

Anemia is defined by the World Health Organization as a decrease in hemoglobin (Hb) value below 13 g/dL in adult men and below 12 or 11 g/dL in non-pregnant or pregnant adult women, respectively [1]. Iron-deficiency anemia (IDA) occurs in 2–5% of adult men and postmenopausal women in the developed world [1,2] and is a very common reason for referral to gastroenterologists. It can be a red-flag sign of asymptomatic colonic and gastric carcinoma, so investigating these conditions is a priority in patients with IDA, especially in the elderly. Malabsorption, most commonly from celiac disease (CD), *Giardia Lamblia* infection, poor dietary intake, use of non-steroidal anti-inflammatory drugs (NSAIDs), and gastric diseases such as corpus atrophic gastritis or *Helicobacter pylori* infection are other common and often overlooked causes of IDA. Multimorbidity and polypharmacological treatments, which might affect the erythropoietic compartment, can play an additional role. Therefore, IDA is often multifactorial and its management is frequently difficult. For this reason, it is recommended that any level of anemia should be investigated so it can be treated, potentially improving the quality of life and outcomes of patients [3]. Recent AGA guidelines recommend a bidirectional endoscopy over no endoscopy in asymptomatic postmenopausal women and men with IDA with moderate-quality evidence [3]. Anemia is a common presentation of CD, and is found in as many as 50% of the patients at the time of the diagnosis [4,5] and in about 60–80% of elderly patients; the incidence of CD in elderly patients (aged >60 years old) has grown during the last decades (from 4% to 19–34%) [6,7]. The need for routine duodenal biopsies in IDA patients, without taking into account their celiac serology results or age, is still debated, mainly because of the implicated costs and the considerable workload on pathologists. The most updated guidelines on the diagnosis and management of CD from the British Society of Gastroenterology recommends, with low quality of evidence, that in individuals undergoing an upper-GI endoscopy of malabsorption, duodenal biopsies should be considered [5], and, if CD is suspected, at least four duodenal biopsy samples should be taken, which is an approach that has been reported to increase the diagnostic yield of CD [8]. The current study aimed to define the yield of routine duodenal biopsies in adult and elderly patients with anemia and without any macroscopic cause of bleeding at the time of the GI endoscopy, and to identify the eventual predictive features of patients in whom duodenal histopathological findings were diagnosed.

## 2. Materials and Methods

### 2.1. Study Population and Design of the Study

This was a single-center, cross-sectional study. All consecutive adult patients (>18 years) who underwent upper-GI endoscopy at Sant’Andrea University Hospital of Rome between 2017 and 2020 and who had anemia as an indication were included. The exclusion criteria were the presence of manifested gastrointestinal bleeding and the absence of duodenal biopsies during upper-GI endoscopy. A total of 7968 upper-GI endoscopies were performed in our Endoscopy Unit in order to identify any indications, out of these 744, of anemia. Among them, 275 patients met one of the exclusion criteria and were thus excluded, and finally, 469 patients were included (Figure 1). The demographic features of the study population were retrospectively collected from the clinical charts (emergency room, inpatients wards, or outpatient clinic) and are reported in Table 1.

Regarding gastrointestinal symptoms, the most common ones were diarrhea, constipation, and abdominal pain. Endoscopic and histological data were retrospectively collected from archived electronic charts. Clinical, endoscopic, and histological features were stratified for the presence or absence of histopathological changes in the duodenal mucosa (DM).

All data from the included patients were anonymized to guarantee the secure processing of sensitive data and collected into a predefined spreadsheet. Written informed consent was obtained from all included patients at the time of endoscopy. The study protocol conformed to the ethical guidelines (approval by the institution’s ethical committee no. 7022/2019, 7 July 2019).

### 2.2. Endoscopy and Histopathology

The upper-GI endoscopy was performed by obtaining multiple biopsy samples, following the standardized updated Sydney protocol, as well as duodenal biopsies (two taken from the duodenal bulb and four taken from the second duodenal portion) for histopathological evaluation. Duodenal biopsies were routinely performed independently of the eventual presence of endoscopically visible duodenal lesions. Gastroscopy was performed using Olympus telescopes GIF-H180, GIF-H185, or GIF-HQ190) by experienced endoscopists with the aid of oral-pharyngeal anesthesia (Xylocaine spray) and/or mild sedation through intravenous infusion of midazolam. In the current study population, the most frequent endoscopic duodenal features were small erosions without signs of bleeding, denting, and decreased duodenal folds. In this study, the main attention was directed to the eventual presence of histopathological changes in duodenal biopsies, therefore the data on the histopathological assessment of gastric biopsies were not reported because they were beyond the aim of this study. The diagnosis of CD was based on the histological examination of duodenal biopsies and defined according to Marsh’s classification [9]. Duodenal intraepithelial lymphocytosis was defined as the increase in intraepithelial lymphocytes in architecturally normal duodenal mucosa. The characteristic appearance of DIL corresponds with Marsh grade 1, i.e., normal or mild increase in the lamina-propria inflammation of the duodenal mucosa, with no crypt hyperplasia or villous atrophy [10]. Chemical duodenitis has been defined when a cluster of patterns was present that exhibited epithelial damage extending up to erosions with consecutive regenerative hyperplasia and damage to capillaries, with edema, hemorrhage, and proliferation of smooth muscles [11]. Lastly, *Giardia Lamblia* infection was diagnosed when the protozoan parasite was present.

### 2.3. Statistic Analysis

The data were expressed as median (range), and/or number/total (percentage, %). In the univariate analysis, the groups were compared by the Chi-squared test for categorical variables or the Mann–Whitney test or Student’s *t*-test for continuous variables. The multivariate analyses were performed by logistic regression (stepwise method, dependent variable: normal duodenal mucosa; independent variables: gender, outpatient/inpatient, age, endoscopic findings in the stomach or the duodenum). Different models were built by using different cut-offs for age (>50, >60, >70, >80 years). Two-tailed *p*-values < 0.05 were considered statistically significant. The statistical analyses were performed by MedCalc^®^ Statistical Software version 19.6.4 (MedCalc Software Ltd., Ostend, Belgium; https://www.medcalc.org; 2021, last access: 8 March 2022).

## 3. Results

Of the 469 included patients, 41 (8.7%) had histopathological changes in the duodenal mucosa. In detail, 12 (2.6%) patients had CD, 26 (5.5%) patients had DIL, 2 (0.4%) had *Giardia Lamblia* infection, and 1 (0.2%) had chemical duodenitis. Figure 2 shows a scatter chart displaying the pathological duodenal findings for each age group of the patients.

### 3.1. Histopathological Findings and Endoscopic Features for Age

Compared to patients with normal duodenal mucosa, patients with duodenal histopathological findings were younger of age (median 47.5 years, range 20–90 vs. median 62, range 18–93, *p* = 0.0005), and showed duodenal endoscopic findings more frequently, such as denting or decreased duodenal folds, (22.5% vs. 6.7%, *p* = 0.0014), while the stomach was endoscopically normal in all cases (0 vs. 14.7%, *p* = 0.0182).

### 3.2. Frequency of Histopathological Duodenal Alterations for Age Groups

The frequency of histopathological findings in the duodenal mucosa showed a significantly decreasing trend in groups with increasing age. In patients aged less than 40–50 years, between 51–60 years, between 61-70 years, between 71–80 years, and over 80 years of age, they were observed in 27%, 20%, 12.5%, 10%, and 2.5%, respectively (chi-squared trend *p* = 0.0010) (Figure 3).

### 3.3. Strength of Association between Endoscopically and Histopathologically Normal Mucosa and Increasing Age

As shown by the logistic-regression analyses, the endoscopically normal duodenal mucosa and ages over 50, 60, and 70 years were significantly associated with normal duodenal-mucosal biopsies at the time of the histopathological assessment. In thegroups with increasing age, the strength of the association increased: patients with anemia aged >60, >70, or >80 years with a normal DM at the time of endoscopy had a three- to four-fold higher probability of having normal duodenal histology (OR 3.5, 95% CI 1.5 to 8.2, OR 3.6, 95% CI 1.6 to 8.5, OR 4.1, 95% CI 1.8 to 9.4) (see Table 2). The independent variables, such as gender, outpatient/inpatient, and gastric endoscopic findings) were not significantly associated. 

## 4. Discussion

Anemia is a common reason for referral to gastroenterologists and is frequently associated with a wide spectrum of gastrointestinal diseases, sometimes resulting in histopathological changes in the duodenal mucosa. The role of performing routine duodenal biopsies during the endoscopic evaluation for the presence of IDA is still debated, because of the implicated costs and the considerable workload on pathologists, but it is increasingly emphasized. The British Society of Gastroenterology guidelines recommended that without manifested bleeding or any other evident cause of IDA, all IDA patients should undergo gastroscopy with duodenal biopsies [12]. The revised guidelines published ten years later recommended that in IDA patients, duodenal biopsies should be performed only in the case of a positive or unperformed celiac serology [13]. The published ACG clinical guidelines for the diagnosis and management of CD recommend routine duodenal biopsies to be performed during gastroscopy when the prevalence of CD is about 5% or higher. Since among patients with IDA the the prevalence of CD reached about 5%, in this clinical scenario, duodenal biopsies and serological testing for tTG antibodies are recommended, in particular in patients aged over 55 years with persistent dyspeptic symptoms despite treatment [14]. The British guidelines concerning the diagnosis and management of CD recommend that in subjects undergoing gastroscopy for anemia, weight loss, or diarrhea, duodenal biopsies should be performed, independently of their CD serology status [5]. Interestingly, routine duodenal biopsies in patients with IDA, regardless of the celiac serology status, or in patients with IDA with negative serology, have been reported to be a cost-effective strategy in patient groups aged >45 years and >65 years [4]_._ Moreover, in accordance with the quality standards of upper-GI endoscopy, in IDA patients, separate biopsies of the gastric antrum and corpus mucosa should be performed, together with duodenal biopsies if the celiac serology is positive or has not been previously tested [15]. In the case of asymptomatic adult patients with IDA and possibile clinical suspicion of CD, the AGA has recently suggested initial serologic testing followed by a duodenal bowel biopsy only if positive, rather than routine duodenal biopsies [3].

CD has been traditionally thought to be more prevalent in children and young adults. In recent years, the incidence of CD diagnosis in elderly subjects has been substantially growing [16]. The prevalence of CD has been estimated to be about 1% in the general population, and the incidence of CD in subjects aged over 65 years has gradually increased from 4% to 19–34% [17]. When present, gastrointestinal symptoms in elderly subjects are typlically mild, making the diagnosis challenging; the most common clinical sign is represented by anemia, mainly due to iron deficiency, which was observed in up to 80% of elderly subjects with CD [13]. The diagnosis of CD in elderly subjects can be delayed for several reasons. The majority of these patients have only mild symptoms and this may be explained by the restricted mucosal extent of the disease; in elderly patients, cognitive impairment may be common, making the definition and awareness of some symptoms more difficult [18]. In elderly subjects, the index of clinical suspicion of CD is frequently low and clinical ivestigations will prevalently be carried out in order to rule out or diagnose more fearful conditions such as colon cancer, which may typically manifest itself with anemia [19]. The current study showed that in anemic patients without endoscopic signs of bleeding, the predictors for histologically normal duodenal mucosa were age and an endoscopically normal duodenal-mucosa appearance. In anemic patients aged over 70, pathological findings in the duodenal mucosa were infrequent (one patient with CD, one with DIL and two with chemical duodenitis). Thus, in these cases, it seems reasonable that the overall clinical–paraclinical context should dictate the decision on an invidual basis of whether duodenal biopsies should be performed or not. One limit of the current study was that the data on the use of drugs, which is common among the elderly and possibly implicated in histopathological alterations of the duodenal mucosa, were not available. Additionally, other anamnestic data such as family history or biochemical data might play a role in selecting patients as candidates for duodenal biopsies with a higher or lower chance oif developing duodenal disease, but unfortunately they were not available in this study. In conclusion, the results of the current study suggest that performing routine duodenal biopsies is questionable in elderly anemic patients, especially in the absence of macroscopic duodenal alterations or other suspicious signs of related intestinal diseases. Further prospective studies are needed to better define predictors for the absence of duodenal histopathological alterations in anemic patients in order to guarantee the care of patients and the high diagnostic yield of upper-GI endoscopy, thereby avoiding useless bioptic sampling. In the meantime, serology may be used to rule out CD.

## Figures and Tables

**Figure 1 diagnostics-12-00678-f001:**
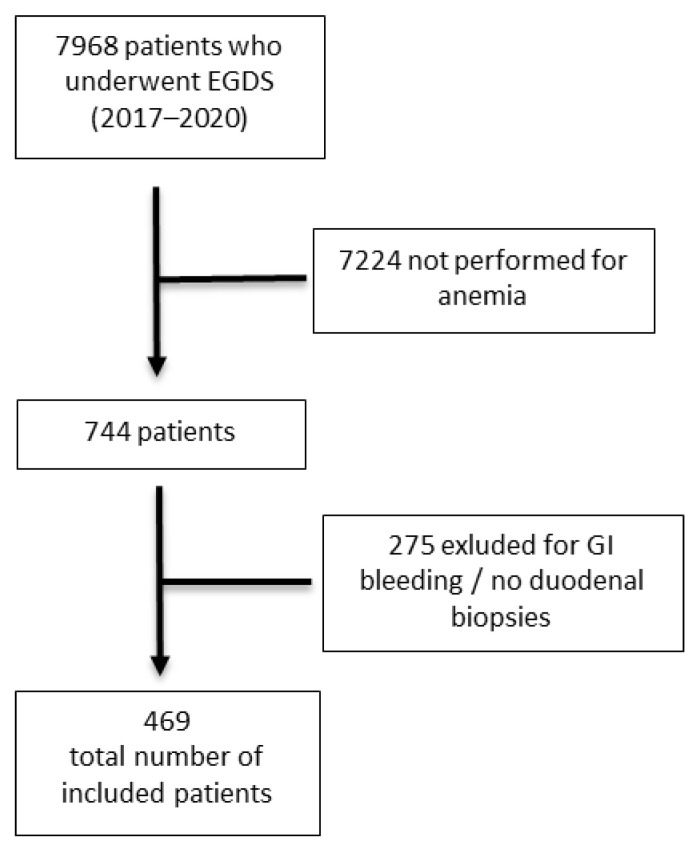
Flowchart. EGDS = esophagealgastroduodenoscopy; GI = gastrointestinal.

**Figure 2 diagnostics-12-00678-f002:**
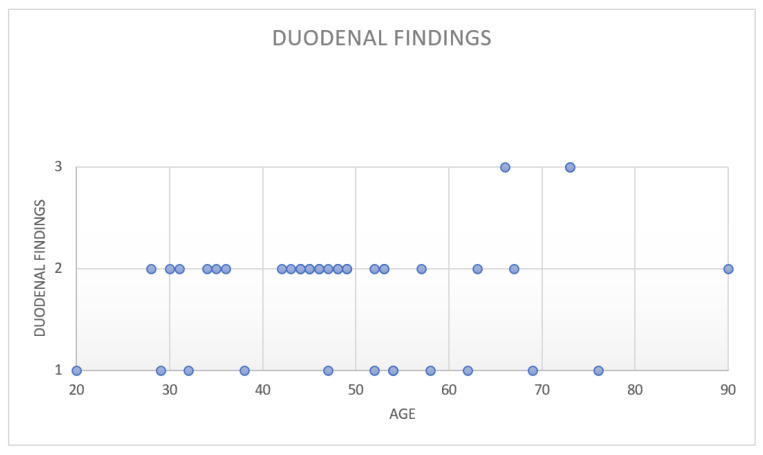
Pathological duodenal findings with respect to age. Legend: 1 = celiac disease, 2 = duodenal intraepithelial lymphocytosis, 3 other findings (*Giardia L* infection, chemical duodenitis).

**Figure 3 diagnostics-12-00678-f003:**
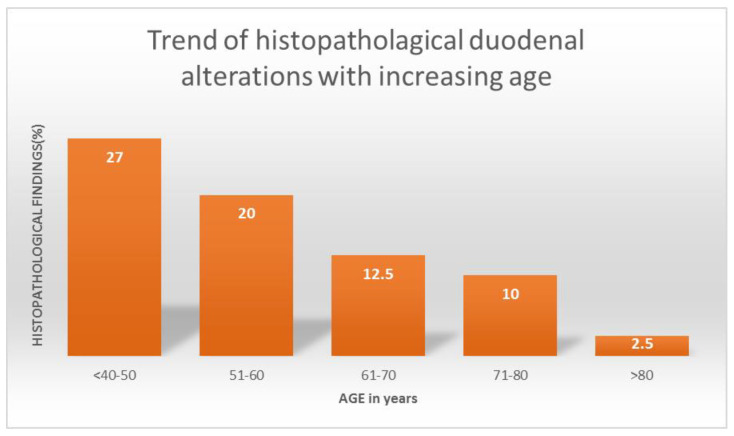
Decreasing frequency of histopathological duodenal alterations with increasing age.

**Table 1 diagnostics-12-00678-t001:** Study population.

	N (469)	%
Females	315	67%
Median age, years	54.7 (18–93)	
Provenience:		
-Outpatients	268	57.1%
-Emergency room	22	4.5%
-Inpatients	119	25.4%
Gastrointestinal symptoms	55	11.7%

**Table 2 diagnostics-12-00678-t002:** Logistic-regression models of features associated with normal histopathological findings of duodenal-mucosa biopsies.

	Odds Ratio	95% CI
Age under 50 years	1	
Age over 50 years	20,565	10,547 to 40,101
Presence of endoscopic findings in duodenum	1	
Absence of endoscopic findings in duodenum	35,426	15,132 to 82,938
Age under 60 years	1	
Age over 60 years	30,187	14,278 to 63,825
Presence of endoscopic findings in duodenum	1	
Absence of endoscopic findings in duodenum	35,007	14,929 to 82,090
Age under 70 years	1	
Age over 70 years	30,184	11,485 to 79,328
Presence of endoscopic findings in duodenum	1	
Absence of endoscopic findings in duodenum	36,409	15,643 to 84,741
Age under 80 years	1	
Age over 80 years	42,501	0.5620 to 321,424
Presence of endoscopic findings in duodenum	1	
Absence of endoscopic findings in duodenum	40,749	17,674 to 93,948

## Data Availability

Authors will provide data on request.

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
