# Peer review of "In Elderly Anemic Patients without Endoscopic Signs of Bleeding Are Duodenal Biopsies Always Necessary to Rule Out Celiac Disease?"

_diagnostics, 2022, doi:10.3390/diagnostics12030678_

Round 1
Reviewer 1 Report
I have read carefully the manuscript sent by you and I consider that it is well structured, the working method is reliable and the discussions are interesting.
The issue raised by this article is whether or not to perform endoscopy in adult or elderly patients with anemia but who do not have visible lesions in the upper digestive tract.
In this context, my opinion is that in the conditions in which the only diagnostic method applied to these patients is endoscopy, the biopsy should be performed. If anamnesis data, objective examination and biological laboratory tests are available, I consider that the decision to perform or not the biopsy is dictated according to the data provided by them.
Therefore, it would have been desirable for the manuscript sent by you to specify in what context the biopsy for this group of patients is not necessarily necessary.
As current guidelines show, consensus on this is hard to come by even digestive endoscopy societies.
Author Response
Reviewer 1
Comments and Suggestions for Authors
I have read carefully the manuscript sent by you and I consider that it is well structured, the working method is reliable and the discussions are interesting.
The issue raised by this article is whether or not to perform endoscopy in adult or elderly patients with anemia but who do not have visible lesions in the upper digestive tract.
In this context, my opinion is that in the conditions in which the only diagnostic method applied to these patients is endoscopy, the biopsy should be performed. If anamnesis data, objective examination and biological laboratory tests are available, I consider that the decision to perform or not the biopsy is dictated according to the data provided by them.
Therefore, it would have been desirable for the manuscript sent by you to specify in what context the biopsy for this group of patients is not necessarily necessary.
As current guidelines show, consensus on this is hard to come by even digestive endoscopy societies.
We thank the Reviewer for the effort to review our manuscript and his valuable comments.
We fully agree with the Reviewer that consensus on this issue on whether to perform duodenal biopsies in elderly anemia patients without bleeding signs is not easy to obtain. We agree further with the reviewers, that since guidelines are not clear on this point, probably the decision of whether to take biopsies or not should be taken on an individual basis taking into consideration all possible risk factors for duodenal disease.
According to our data, the context in which the diagnostic yield of duodenal biopsies dramatically decreases is mainly age, while other possible predictors, such as age, to be an outpatient or an inpatient or to have endoscopic findings in the stomach did not seem to play a role. This has been reported in the Results section (see page 5, paragraph 3.3). We are aware that other factors or variables which were not available in this study, might eventually have a role in this context. One of these probably may be represented by drugs. To meet this criticism, we have now added a paragraph on the limits of the study discussing this point (see page 6-7, Discussion section).
Reviewer 2 Report
Authors describe that routine duodenal biopsy may be unnecessary in elderly anemic patients. This article is well-analyzed. However, the clinical impact remains obscure.
However, I would like to suggest some issues of this article with several comments and criticisms as following.
Major.
- Inclusion criteria:
What was the detailed criteria for performing multiple duodenal biopsies in your facility? Please clarify this including target diseases (CD, inflammatory bowel disease, eosinophilic enterocolitis, etc.)
- If this study was a retrospective fashion, authors need to mention it.
- What were outcomes measures in this study?
Since endoscopic duodenal findings are commonly The major significance of biopsies remains unclear.
Of the 469 included patients, 41 (8,7%) had histopathological changes in the duodenal 116 mucosa. In detail, 12 (2,6%) patients had CD, 26 (5.5%) patients had DIL, 2 (0.4%) had Giardia Lamblia infection, and 1 (0.2%) had chemical duodenitis.
I wonder if duodenal biopsies aimed to capture these diseases in adult and elderly patients without bleeding anemia.
To make the title more impressive, the target disease of biopsies should be involved.
- Considering the increased medications (Antithrombotics, Nsaids etc.) in elderly patients, these drugs may be factors associated with abnormal duodenal mucosa. It seems a crucial point how duodenal biopsies can contribute in elderly patients with these medications uptake. Authors need to analyze these relationships.
Minor.
- Abstract: Authors should the full word when using abbreviations of CD, DIL. (Exam. celiac disease (CD))
- In line 37, shouldn't "th" be "the"?
- "Duodenal intraepithelial lympthocytosis" in lines 95-96 should be corrected to "Duodenal intraepithelial lympthocytosis(DIL)"?
- In line 139, shouldn't "Ainge" be "Age"?

Author Response
Reviewer 2
Comments and Suggestions for Authors
Authors describe that routine duodenal biopsy may be unnecessary in elderly anemic patients. This article is well-analyzed. However, the clinical impact remains obscure.
We thank the Reviewer for the effort to review our manuscript and his valuable comments.
In our opinion, the clinical impact of the paper lies in the fact, that in elderly patients, duodenal biopsies may be reasonably waived because the adjunctive diagnostic yield of eventual duodenal alterations seems, mainly celiac disease and duodenal intraepithelial lymphocytosis without atrophy seem to be very low. Duodenal biopsies are routinely performed in patients with iron-deficiency anemia without bleeding, mainly to exclude celiac disease, very common in Italy and many other European countries. Guidelines on this point are not clear and the diagnostic yield of celiac disease in elderly patients is not so high, albeit the prevalence in this age group has been increasing in the last years.
However, I would like to suggest some issues of this article with several comments and criticisms as following.
Major.
- Inclusion criteria:
What was the detailed criteria for performing multiple duodenal biopsies in your facility? Please clarify this including target diseases (CD, inflammatory bowel disease, eosinophilic enterocolitis, etc.)
We thank the Reviewer for this interesting comment. In our GI endoscopy unit, besides in target diseases, multiple duodenal biopsies (two from the duodenal bulb and four from the second duodenal portion) for histopathological evaluation are routinely obtained in patients with anemia without bleeding lesions, except for patients with contraindications, as blood clotting disorders.
So, duodenal biopsies are routinely obtained independently of the eventual presence of duodenal lesions or suspicion of duodenal diseases. This routine endoscopy approach in our unit is related to the fact that our teaching hospital is a referral center for iron-deficiency anemia and disorders related to iron-malabsorption and autoimmune diseases. This point has now been clarified in the Methods sections (see page 3, paragraph 2.2).
- If this study was a retrospective fashion, authors need to mention it.
We thank the Reviewer also for this important comment. This was a single-center cross-sectional study on consecutive adult patients undergoing upper GI endoscopy for anemia. Data on demographics, endoscopy, and histology were retrospectively collected from clinical charts and analyzed. This has now been better clarified in the Methods section (see page 2-3, paragraph 2.1)
- What were outcomes measures in this study?
Since endoscopic duodenal findings are commonly The major significance of biopsies remains unclear.
We thank the Reviewer also for this pertinent comment. The outcome measure of the current study was to assess the diagnostic yield of routine duodenal biopsies in elderly patients with anemia without bleeding at gastroscopy. Duodenal biopsies are routinely performed in patients with iron-deficiency anemia without bleeding, mainly to exclude celiac disease, very common in Italy and many other European countries. Guidelines on this point are not clear and the diagnostic yield of celiac disease in elderly patients is not so high, albeit the prevalence in this age group has been increasing in the last years. In this study, the presence of common duodenal lesions related to manifest gastrointestinal bleeding (diffuse erosions with signs of bleeding, peptic ulcer) was excluded because only iron deficiency anemia patients without bleeding signs were included. The main endoscopic findings in this study were denting a and rarefaction of duodenal folds and were present in 29% of patients, but significantly more common in patients with duodenal histopathological findings and of younger age.
Of the 469 included patients, 41 (8,7%) had histopathological changes in the duodenal mucosa. In detail, 12 (2,6%) patients had CD, 26 (5.5%) patients had DIL, 2 (0.4%) had Giardia Lamblia infection, and 1 (0.2%) had chemical duodenitis. I wonder if duodenal biopsies aimed to capture these diseases in adult and elderly patients without bleeding anemia. To make the title more impressive, the target disease of biopsies should be involved.
We thank the Reviewer for this good suggestion. What we had in mind when we designed the study was essentially the diagnostic yield of celiac disease and duodenal intraepithelial lymphocytosis (Marsh 1), which indeed were the most frequently obtained histopathological changes. As suggested, we have now modified the title to include the target disease of duodenal biopsies (see Title, page 1).
- Considering the increased medications (Antithrombotics, Nsaids etc.) in elderly patients, these drugs may be factors associated with abnormal duodenal mucosa. It seems a crucial point how duodenal biopsies can contribute in elderly patients with these medications uptake. Authors need to analyze these relationships.
We thank the Reviewer for this important comment with which we fully agree. In this study, we found only one patient with chemical, probably drug-related duodenitis (male, 73 years) and 26 patients with duodenal intraepithelial lymphocytosis of whom only 4 patients were aged older than 60 years. Of course, in these cases, the role of drugs cannot be excluded. At the same time, as commonly the use of drugs increases with age, if mainly related to drugs, these duodenal alterations should have been expected to be more frequent in the elderly age group. In the current study, the duodenal intraepithelial lymphocytosis was more frequent in the younger age groups. Unfortunately, data regarding the use of drugs was not available in this study, but we agree with the Reviewer that this item deserves further evaluation. However, we have now discussed this point as a limit of the study (see page 6, Discussion section).
Minor.
- Abstract: Authors should the full word when using abbreviations of CD, DIL. (Exam. celiac disease (CD))
Thank you for this comment, the full expression “celiac disease (CD)” and duodenal intraepithelial lymphocytosis (DIL) has now been included (see page 1, Abstract).
- In line 37, shouldn't "th" be "the"?
Thanks, we have corrected this (page 1, Introduction).
- "Duodenal intraepithelial lympthocytosis" in lines 95-96 should be corrected to "Duodenal intraepithelial lympthocytosis(DIL)"?
Thank you, we have corrected also this error (page 3, paragraph 2.2)
- In line 139, shouldn't "Ainge" be "Age"?
Yes, of course it should be “Age”. We apologize for this error (page 5, title of paragraph 3.3)
Round 2
Reviewer 1 Report
The team did an excellent job of improving the scientific content of the manuscript and gave a better meaning to the purpose of its publication. Although, I remain of the opinion that there are still many controversies regarding the performance of duodenal biopsy in patients with anemia, I consider that the overall clinical-paraclinical context dictates the conduct in such cases and in this context the decision is that of the doctor performing endosocopy.
Author Response
We thank the Reviewer again for his comments with which we fully agree. We added a sentence in the discussion section to strain the point, that, at least until guidelines on elderly not-bleeding patients do not clearly include the need or not of duodenal biopsies the ultimate decision will be taken by the GI endoscopist on an individual basis.
Reviewer 2 Report
Authors need to add a conclusion how celiac disease could be ruled out.
Other revisions are satisfied.
Author Response
We thank the Reviewer again for his revision and comments. As requested, a sentence was added in the conclusion that serology may be used to rule out CD.